# Acidic Gases Solubility in Bis(2-Ethylhexyl) Sulfosuccinate Based Ionic Liquids Using the Predictive Thermodynamic Model

**DOI:** 10.3390/membranes10120429

**Published:** 2020-12-16

**Authors:** Amal Mechergui, Alsu I. Akhmetshina, Olga V. Kazarina, Maria E. Atlaskina, Anton N. Petukhov, Ilya V. Vorotyntsev

**Affiliations:** 1Laboratory of Membrane and Catalytic Processes, Nizhny Novgorod State Technical University n.a. R.E. Alekseev, 24 Minina str., Nizhny Novgorod 603950, Russia; amalfatroucha@gmail.com (A.M.); aai-89@mail.ru (A.I.A.); olga_kazarina@list.ru (O.V.K.); m.e_salnikova@mail.ru (M.E.A.); fox_off@mail.ru (A.N.P.); 2SMART PolyMaT Key Laboratory, D. Mendeleev University of Chemical Technology of Russia, Miusskaya Sq. 9, Moscow 125047, Russia

**Keywords:** gas mixtures, ionic liquid membrane, hydrogen sulfide, carbon dioxide, sorption, thermodynamic

## Abstract

To properly design ionic liquids (ILs) adopted for gases separation uses, a knowledge of ILs thermodynamic properties as well their solubilities with the gases is essential. In the present article, solubilities of CO_2_ and H_2_S in bis(2-Ethylhexyl)sulfosuccinate based ILs were predicted using the conductor like screening model for real solvents COSMO-RS. According to COSMO-RS calculations, the influence of the cation change was extensively analyzed. The obtained data are used for the prediction of adequate solvent candidates. Moreover, to understand the intrinsic behavior of gases solubility the free volume of the chosen ILs and their molecular interactions with respectively CO_2_ and H_2_S were computed. The results suggest that hydrogen bonding interactions in ILs and between ILs and the gases have a pivotal influence on the solubility.

## 1. Introduction

Among the current environmental issues, the removal of acidic gases such as carbon dioxide and hydrogen sulfide has received great attention due to the necessity of preventing the greenhouse effect responsible for irreversible climate changes and negative impact on the biosphere. Nowadays, conventional technology for acidic gases removal in the industry is based on the chemisorption process by an aqueous alkanolamine solution, unfortunately not free of some drawbacks including high energy demand during the regeneration of the liquid, solvent loss and degradation [1]. In this regard, the uprising trend of replacing conventional acidic gases separation techniques by the economically viable and energy-effective membrane or membrane assisted technology has been gaining much interest over the last years, also by authors [2,3,4,5,6]. In the field of membrane gas separations, a novel class of membranes composed of ionic liquids (ILs) or their polymeric derivatives has exhibited an excellent promise for acidic gases separations and has overcome the trade-off outlined in the Robeson upper bound [7,8,9]. The utilization of ILs for gas capture purposes was first investigated by Blanchard et al. [10]. Research efforts [11,12] in this area have been increasing. Designing task-specific ionic liquids by the chemical alteration of the cation or anion such as tethering a specific functional group [13,14,15,16,17,18] has attracted great interest in the last few years. The possible combinations between the already existing cations and anions result in a large variation of ionic liquids with unique properties. Due to their high cost and limited number, the experimental investigation of all possibly existing ILs has become a challenging task. Therefore, a prior screening and design of ionic liquids with different gases absorption capacities using a reliable computational method [19,20,21,22,23] would be of great value for subsequent experimental work. Gaining much attention over the last few years, the conductor-like screening model for real solvents (COSMO-RS) [24,25,26] has successfully predicted CO_2_ solubility in a wide range of ionic liquids. This method is based on the quantum chemical description of individual molecules. It only requires universal parameters and element-specific parameters. Thus, COSMO-RS can be considered as an adequate tool for the predictions of ILs thermodynamic properties, their activity coefficients and their Henry’s law constants [26,27,28]. Additionally, the solubility characteristics of some macromolecules and polymers [29,30] in ILs have also been modelled by COSMO-RS. Overall, COSMO-RS can be applied for the calculation of liquid–liquid equilibria and solvent prescreening. In the present work, the solubility of CO_2_ and H_2_S in four different ILs (Table 1) has been predicted by COSMO-RS. The effect of the type of the cation and its structural variations on gases solubilities have been examined systematically using extensive computational screening considering the ILs molecular structures.

## 2. Theory

### 2.1. Theory of Calculation Methods

Conductor like screening model for real solvents COSMO-RS is a new method for the prediction of thermophysical and chemical properties of fluids and liquid mixtures based on unimolecular quantum chemical calculations [31]. The full description of the methodology can be found elsewhere, and only major features needed for understanding the analysis and the discussion of the obtained results are highlighted here. Each COSMO-RS calculation includes two steps. The first step is the quantum chemical COSMO/DFT calculations of the chosen molecules. During this stage, each entity is individually inserted in a perfect conductor where a cavity is constructed around the molecule so it induces a charge distribution in the discrete surface (the cavity) between the counterion and the conductor. These charges are called screening charges, and the surface density *σ* usually represents them. This charge distribution is considered as the most important molecular descriptor. The obtained screening charge density of individual molecules are converted into the probability distribution function (histogram) *p*′(*σ*) or *σ*-profile. The *σ*-profile of a mixture *p*(*σ*) is built by adding the *p*′(*σ*) of the already screened molecules weighted by their mole fractions in the mixture [32]. Having the *σ*-profile, the chemical potential is calculated by solving couples of non-linear equations. Finally, the obtained chemical potential is used to calculate thermodynamical properties such as activity coefficients, Henry’s Law constants, solubilities of liquids and gases in IL etc. The main working equations are given by

The *σ*-profile *p*(*σ*):(1)$$p\left(\sigma \right)=\underset{i}{\sum }x_{i}p_{i}^{\prime }\left(\sigma \right)$$

The chemical potential: (2)$$µ\left(\sigma \right)=\;-KT\ln\int p\left(\sigma \prime \right)\exp\left\{-\frac{E_{\mathrm{int}}\left(\sigma ,\;\sigma \prime \right)-µ\left(\sigma \prime \right)}{KT}\right\}d\sigma \prime$$

The activity coefficient:(3)$$\gamma =\exp\left(\frac{\mu -\mu^{0}}{\mathrm{RT}}\right)$$
where: *x_i_* is the molar fraction of a molecule within a mixture, *E*_int_ is the interactions energy, *T* is the temperature, *K* is Boltzmann constant and µ^0^ is the chemical potential at pure state.

### 2.2. Computational Details 

Densities, molar volumes and COSMO volumes were obtained for each of the four cation−anions using the COSMOtherm software (Version 3.4). When available, the considered species were directly used from the COSMObase libraries derived from the software provider (COSMOlogic GmbH, Leverkusen, Germany). For those cation and anion species not available within the COSMObases, COSMO files were developed. Optimized structures of cations and anions were designed with TURBOMOLE [32], using the triple-ζ valence potential (TZVP) basis set [33] with the Becke and Perdew (BP) functional [34,35] at the density functional theory (DFT) level. All COSMO calculations were performed at the TZVP level of theory, consistent with other published works on ILs that have utilized COSMOtherm. Conductor like screening model for real solvents COSMO-RS is a new method for the prediction of thermophysical and chemical properties of fluids and liquid mixtures based on unimolecular.

### 2.3. Computational of Fractional Free Volume

Initially, the free volume was defined as the difference between the molar volume (Vm), and the Van der Waals volume (VVdW) calculated from the Van der Waals radii (rvdW) using the methods and values set proposed by Bondi.
$$V_{f}=V_{m}-1.3V_{vdW}$$

Lately, it has been proven that COSMO radii calculated from the COSMOtherm software are 1.17 times rvdW; therefore, the VVdW can be correlated to COSMO volume VCOSMO which presents the volume enclosed by the COSMO surface or the accessible area of the solvent. As found in the literature and determined by Bara et al., VCOSMO ≈ 1.3 VVdW thus:$$V_{f}=V_{m}-V_{COSMO}$$

The fractional free volume (*FFV*) is defined as the ratio of the space to the occupied space in the considered ILs:$$FFV=\frac{V_{m}-V_{COSMO}}{V_{m}}$$

## 3. Results and Discussions 

### 3.1. σ-Profiles of the Ionic Liquids

The charge distribution sigma *σ* on the molecular surface can be easily visualized in the histogram function *σ*-profile for the investigated ionic liquids which all was with docusate anion ([doc], see Table 2), presented in Figure 1. These graphs of the ILs may be analyzed to understand the ions pair’s effects on the ILs properties. Similar to any other ionic liquid, the sigma profiles of [doc] based ILs are qualitatively divided into three regions: hydrogen bond acceptor region for *σ* > 1 e/nm^2^, hydrogen bond donor region for *σ* < −1 e/nm^2^ and the neutral region for the zone in between. The peak around 1.68 e/nm^2^ at the high polarity zone is assigned to the polar segment in [doc] anion as the distribution charge densities around 0 e/nm^2^ correspond to the alkyl groups on the anion. In the negative region, we observe peaks at values lower than -1 e/nm^2^ which are related to the hydrogen atoms on the [bmim], [mim], [empyrr] and [hpyr] cations capable of acting as potential hydrogen bond donors. Similar to the [doc] anion the sigma distribution between 1 e/nm^2^ and −1 e/nm^2^ are assigned to the nonpolar alkyl groups on the imidazolium, pyrrolidinium and pyridinium rings.

### 3.2. Screening Charges Densities

DFT calculations were carried out to visualize the structural variations of the cations on the gases solubility of the studied ILs. As shown in Figure 2, the distribution of the charge on the surface of cations varies considerably for different heterocycles. Two acidic protons N(3)-H and C(2)-H were observed in the structure of [mim] cation N(3)-H proton acidity was more pronounced. In the case of [bmim] cation, the most acidic site was located in C(2)-H bond. The positive charge in the [empyrr] cation was localized in the nitrogen atom, however, on the COSMO-surface the charge was found to be equally distributed because of the presence of alkyl substituents on the nitrogen atom. Finally, C(2)-H and C(6)-H protons have shown higher acidity than other ones within [hpyr] heterocycle.

### 3.3. Densities and Volumetric Effects

The next step was to evaluate the densities of [doc] ILs using the COSMO-RS model. All the predicted densities values are illustrated in Figure 3, with a comparison to the experimental data of the ILs.

An excellent linear relationship has been found between the experimental and the calculated densities for all the designed ILs. The linear regression fits represent excellent correlation coefficients *R* = 1 as for the slopes they are presented in Table 2. The root means square deviations (rmsd) of [bmim][doc], [mim][doc], [empyrr][doc], [hpyr][doc] and some other common ILs are given in Table 3. Although COSMO-RS calculated densities of the organic solvents and the common ILs presented rmsd <2% [31], docusate based ILs had higher values. An explanation of such deviation is not evident yet, but this can be related to the structure and the nature of the anion used in this study. It is important to note that the prediction of the ILs densities follows the experimental tendencies related to the cation change effects and the temperature increase [3,36]. Thus, similar to the experimental results, in the case of [mim][doc] and [bmim][doc] the addition of the alkyl chain on the imidazolium ring resulted in higher densities.

Besides, the molar liquid volume was also computed by the model and compared to the experimental presenting an excellent correspondence with each other. Similar to the density, the calculated results computed the same tendency as the experimental results. The densities of ionic liquids were found to increase with the molecular volume, but this is not a general rule. It may prove that the intermolecular interactions in the ILs also govern the physicochemical properties of the ILs.

### 3.4. Henry’s Law Predictions

In Table 4 the predicted Henry’s law constants of CO_2_ and H_2_S in mim[doc], bmim[doc], hpyrr[doc] and empyrr[doc] are reported.

According to the computed results, structural modifications on the imidazolium ring by the addition of the butyl chain on [mim] cation have increased both CO_2_ and H_2_S dissolution in [bmim][doc]. This observation is in good agreement with K.Z. Sumon et al. (2011) [37] work where the authors have proposed that the increase of the alkyl chain length in ring-precursor enhances CO_2_ solubility within a given IL. Additionally, similar to K.Z. Sumon et al. work, the solubility of both gases has remarkably improved (Henry’s law constants decreased) when the cation family has been modified; for both CO_2_ and H_2_S the solubility is ranking in the order empyrr[doc] > bmim[doc] > hpyrr[doc] > mim[doc]. 

Further, bmim[doc] predicted data were compared to those of bmim[BF_4_] and bmim[acetate]. As suggested in the literature [38,39,40], the anion change has a more pronounced effect on gases solubility. The variation in Henry’s law constants when replacing [doc] anion by [BF_4_] and [acetate] anions are larger. The utilization of the docusate anion has enhanced the bmim based ILs sorption capacities for CO_2_. The bulkier nature and the larger molecular weight of the anion have contributed to this augmentation.

The predicted data has shown that docusate based ILs have a higher affinity towards H_2_S than CO_2_. The effects of the alkyl chain modification on the imidazolium-based ILs and the variation of the cation family have been accurately predicted. However, the order of solubility of CO_2_ and H_2_S in the ILs computed by COSMOtherm is different from the experimental.

The experimental data on Henry’s constants are given also in [3], where it is between two and four times less for the cations [mim] and [bmim], respectively. Such a large difference is apparently realized due to the fact that the experimental data for these substances are given with a sufficiently high water content of 0.3722 wt% and 0.4835 wt%.

According to COSMO-RS calculations, empyrr[doc] has exhibited the highest CO_2_ and H_2_S solubilities, but the experiments suggested that bmim[doc] has the highest solvation capacities for both gases.

### 3.5. Interpretation of Molecular Interactions with Sigma Profiles and Sigma Potentials

In this section, we explore the above trends through the properties of ILs and gas-liquid interactions. As mentioned before, the observed trends of solubilities are results of the molecular interactions first between the anions and the cations in the ILs and secondly between the ILs and the gases upon dissolution. The sigma-profiles of the ILs are used to visualize the effect of structural variations on CO_2_ and H_2_S solubilities. As for the corresponding *σ*-potentials, they can be utilized to ascertain the affinity of the ILs towards the gases.

Almost all the *σ*-profiles of the four ILs extended between −25 e/nm^2^ and 20 e/nm^2^. All the ILs sigma-profiles have exhibited similarities in the region around 5–20 e/nm^2^ owing to the docusate presence being the commonly used anion in the designed ILs. The main difference between the histograms of the ILs is observed in the negative region or precisely in the hydrogen bonding donor region. As concluded from Henry’s law constants values the docusate based ILs have a better affinity towards H_2_S than CO_2_. When comparing the *σ*-profiles of both gases, it is obvious that CO_2_ is more localized in the neutral region whereas H_2_S histogram extends beyond both negative and positive regions proving the stronger ability of H_2_S to interact with the ILs via hydrogen bonding. bmim[doc] has exhibited higher solubilities for both gases than mim[doc]. Although bmim[doc] *σ*-profile presents more nonpolar surfaces around *σ* 0 e/nm^2^ (which is favorable for CO_2_ solubility in the IL), the histogram of mim[doc] extends further in the negative region suggesting a stronger hydrogen bonding donor ability than the rest of the ILs. The fact that mim[doc] exhibited lower gases solubilities despite having more negative surface pieces ready to interact with CO_2_ and H_2_S is due to the protic nature of [mim] cation containing the weak protonated nitrogen in the imidazolium ring resulting in strong hydrogen bonding interactions between the [mim] cation and the docusate anion and in different arrangements of them at the molecular level preventing further interactions between the gases and the ions and significantly reducing IL sorption capacity.

### 3.6. Free Volume Effects and Interactions Enthalpies

Many works have used the molar volume of an IL as a tool for understanding the solubility of a gas [33]. Some predictive models have correlated Henry’s law constants of different gases with the molar of a solvent through its solubility parameters. The scaled particle theory [41] has proposed that Henry’s law constant should be decreased when increasing the molar volume of ILs. Although in our case, the relation between Henry’s law constants and molar volumes didn’t match the correlation mentioned above; [empyrr][doc] with the highest CO_2_ and H_2_S solubilities (the lowest Henry’s law constants) has lower molar volume than [hpyr][doc] and [bmim][doc]. Instead of correlating Henry’s law constants of CO_2_ and H_2_S in the ILs with their molar volumes, Bara et al., [42] proposed that the solubility of a gas can be related to the free volume or the fractional free volume of the IL. The free volume V_f_ is the growing void space that might exist within the IL.

The obtained free volumes values of docusate based ILs, presented in Table 5, were much higher than those reported in previous works [35,36]. As we mentioned before, due to the nature of the used anion it is much bulkier and delocalized than the commonly used anions and the ones available in COSMOtherm database, and probably some future correction should be included to the software for better predictions when treating similar ionic species. It can be seen from the obtained results that the addition of the butyl to the [mim] cation FFV of [bmim][doc] have increased favoring better solubility of gases in the mentioned IL. Otherwise, the FFV tends almost to follow the same trend as the Henry law’s constant, [empyrr][doc] with the lowest Henry law’s constant for both CO_2_ and H_2_S having had the highest FFV. From the observation of the obtained data, it can be confirmed that there is a link between the molar volume, the free volume and the gases solubilities within the ILs. It can be concluded that higher CO_2_ and H_2_S solubilities can be achieved when the molar volume of the IL is minimized while the free volume is increased as in the case of [empyrr][doc].

Using the COSMO-RS model excess enthalpies of gas-IL mixtures can be predicted as the sum of different contributions of each component of the mixture as follow
$$H_{m}^{E}=H_{m}^{E}\left(misfit\right)+H_{m}^{E}\left(H-bonding\right)+H_{m}^{E}\left(VdW\right)$$

According to the equation above, the predicted solubilities of CO_2_ and H_2_S can be analyzed in terms of different gas-solvent interactions contributions to the $H_{m}^{E}$ values of CO_2_-IL and H_2_S-IL systems reported in Table 6.

The highest solubilities of H_2_S and CO_2_ by docusate based ILs are associated with higher exothermicity.

As seen from the obtained energies, CO_2_ solubility within all the prepared ILs is generated by the Van der Waals forces, misfit representing electrostatic interactions are repulsive for both CO_2_ and H_2_S and play a secondary role as for the hydrogen bonding and they are mainly absent for CO_2_ sorption in the four ILs. In the case of H_2_S, the three types of interactions contribute to the solubility of the gas. As the dominant contributions, the Van der Waals interactions are responsible for the exothermicity of CO_2_-IL and H_2_S-IL mixtures and the increasing solubilities of both gases within the designed ILs. 

## 4. Conclusions

Several structural and physical parameters of the four ILs were predicted using the COSMO-RS model for better understanding of the influence of the cation change. As expected, the lowest fractional free volume values have corresponded to the highest Henry’s constants. The predicted data suggested that empyrr[doc] has the highest H_2_S and CO_2_ absorption capacity. This observation can be interpreted by the weaker interactions between the cation and the anion resulting in more available void space within the IL allowing the gases to better with the ionic species in the solvent. In summary, we have proved the applicability of the approach based on the introduction of sterically hindered moieties along with polar functional groups in the anion structure towards the development of H_2_S and CO_2_-selective absorbents.

## Figures and Tables

**Figure 1 membranes-10-00429-f001:**
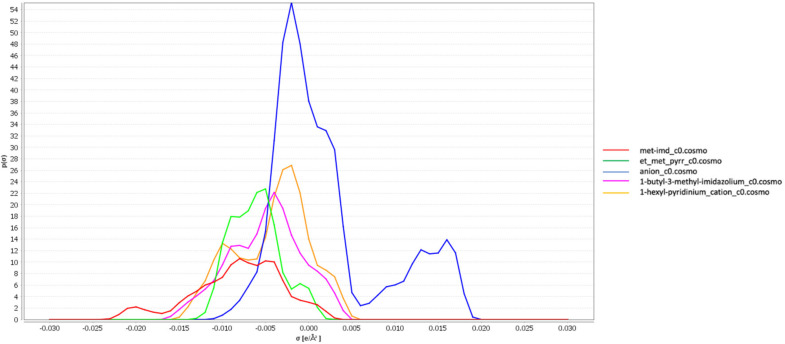
*σ*-profiles of docusate based ionic liquids.

**Figure 2 membranes-10-00429-f002:**
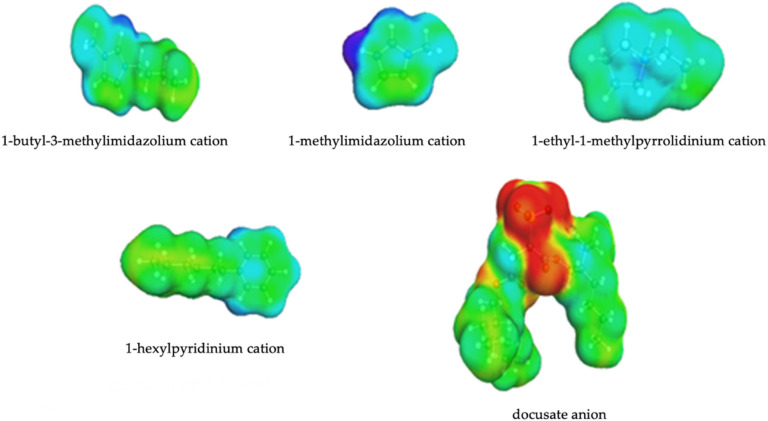
Screening charge *σ* of docusate based ionic liquids.

**Figure 3 membranes-10-00429-f003:**
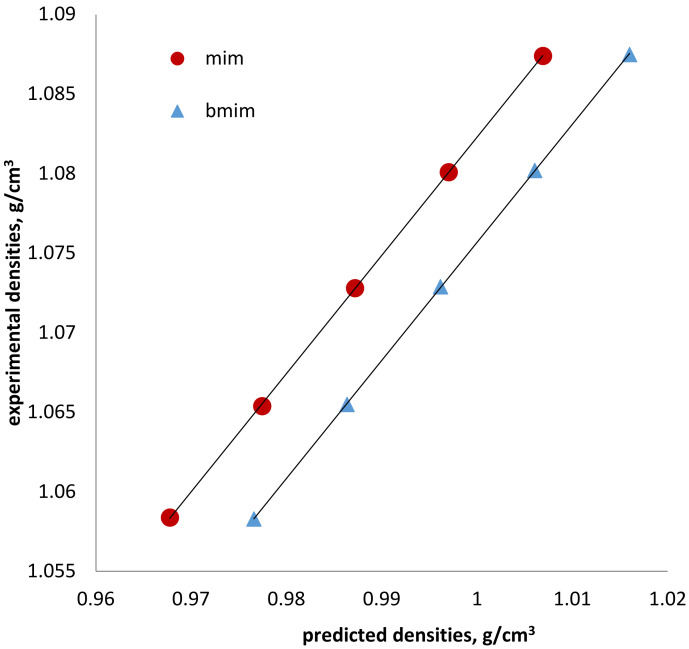
Comparison of experimental and COSMO-RS predicted results of densities (g/cm^3^) for docusate based ionic liquids.

**Table 1 membranes-10-00429-t001:** The investigated cations and anion of ionic liquids.

Abbreviation	Name	Structure	Molar mass/g/mol
[bmim]	1-butyl-3-methylimicazolium cation	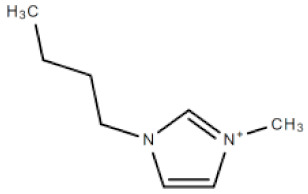	139.17
[mim]	1-methylimidazolium cation	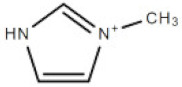	83.10
[empyrr]	1-ethyl-1-methylpyrrolidinium cation	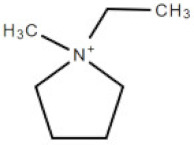	114.21
[hpyr]	1-hexylpyridinium cation	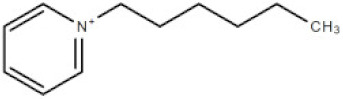	164.21
[doc]	docusate anion (bis(2-ethylhexyl) sulfosuccinate, dioctyl sulfosuccinate (DOSS))	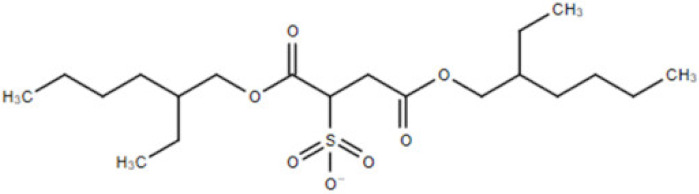	421.57

**Table 2 membranes-10-00429-t002:** The linear regression fits of ionic liquids.

ILs	Linear Regression Fit	Correlation Coefficient, R
[mim][doc]	***ρ***experimental= 0.743***ρ***predicted + 0.3393	1
[bmim][doc]	***ρ***experimental= 0.7418***ρ***predicted + 0.3338	1

**Table 3 membranes-10-00429-t003:** COSMO-RS predictions of densities ***ρ*** and molar volumes (*V_m_*) @ 298.15 K.

ILs	*ρ* (g/cm^3^)		Dev., %	*V_m_* (cm^3^/mol)		Dev., %	Reference
	Experim.	Predict.		Experim.	Predict.		
[mim][doc]	1.084	1.017	6.59	456.587	471.762	3.32	[31]
[bmim][doc]	1.084	1.026	5.63	506.637	546.376	7.84	[31]
[hpyrr][doc]	-	1.013	-	-	575.931	-	This work
[empyrr][doc]	-	1.069	-	-	528.870	-	This work
[bmim][BF4]	1.205	1.194	0.90	187.824	189.028	0.64	[32]
[bmim][CF_3_SO_3_]	1.301	1.319	1.40	220.934	218.526	1.09	[32]
[hxmim][Tf_2_N]	1.370	1.395	1.80	326.886	320.866	1.84	[32]
[emim][Tf_2_N]	1.521	1.531	0.65	291.970	288.358	1.24	[32]

**Table 4 membranes-10-00429-t004:** Predicted Henry’s law constants of CO_2_ and H_2_S in the ionic liquids (ILs).

ILs	*H(bar)*				
	303.15 K	313.15 K	323.15 K	333.15 K	343.15 K
**CO_2_**					
mim[doc]	62.694	75.498	89.712	105.323	122.303
bmim[doc]	45.864	55.511	66.304	78.252	91.351
hpyrr[doc]	45.202	54.566	65.028	76.598	89.276
empyrr[doc]	44.107	53.610	64.301	76.199	89.309
**H_2_S**					
mim[doc]	0.721	1.0254	1.416	1.908	2.513
bmim[doc]	0.523	0.7545	1.058	1.445	1.926
hpyrr[doc]	0.528	0.760	1.061	1.445	1.923
empyrr[doc]	0.461	0.673	0.953	1.315	1.771

**Table 5 membranes-10-00429-t005:** Predictions of free volumes V_f_ and fractional free volumes FFV of ILs @ 298K.

ILs	V_f_ (cm^3^/mol)	FFV
mim[doc]	252.1296811	0.536972372
bmim[doc]	320.5894064	0.589633588
hpyrr[doc]	338.0120416	0.589782822
empyrr[doc]	311.7823921	0.592451172

**Table 6 membranes-10-00429-t006:** The predicted interaction energy contributions of the ILs @ 298.15 K.

ILs	Misfit Interactions Energy Kcal/mol		H-Bonding Interactions Energy Kcal/mol		VdW Interactions Energy Kcal/mol		Total $H_{m}^{E}$ Kcal/mol	
	CO_2_	H_2_S	CO_2_	H_2_S	CO_2_	H_2_S	CO_2_	H_2_S
mim[doc]	1.02556	1.11703	0	−0.6868	−2.89609	−3.65569	−1.87053	−2.08448
bmim[doc]	1.03132	1.12246	0	−0.72946	−2.88958	−3.64518	−1.85827	−2.1119
hpyrr[doc]	1.02511	1.12872	0	−0.71923	−2.89069	−3.65656	−1.8658	−2.10608
empyrr[doc]	1.06730	1.12549	0	−0.76074	−2.86842	−3.60433	−1.80113	−2.09859

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
