# Peer review of "Acidic Gases Solubility in Bis(2-Ethylhexyl) Sulfosuccinate Based Ionic Liquids Using the Predictive Thermodynamic Model"

_membranes, 2020, doi:10.3390/membranes10120429_

Round 1

Reviewer 1 Report

In this manuscript, authors calculated the solubilities of CO2 and H2S in ionic liquids based on COSMO-RS. The results indicate that hydrogen bonding interaction between ionic liquid and gases plays a key role in solubility. It has been known that the gas solubility is crucial for understanding gas removal behavior. The present approach provides new insight to understand such behavior and find proper candidates in an economical way. The manuscript is logically written and data support author’s conclusion appropriately. Therefore, I recommend the manuscript to be publish after minor revisions.

Minor questions

  1. Please explain why authors chose CO2 and H2S and four ionic liquids among other materials. Can author apply the current approach to other materials?
  2. It will be much better to add some more previous references related to the present work.
  3. All the figures looks to blurry. Please replace them with high quality figures.

Author Response

1- Please explain why authors chose CO2 and H2S and four ionic liquids among other materials. Can author apply the current approach to other materials?

Currently we are working in area of separation of acid gases from nitrogen or methane. These to media is very important from the practical and ecological point of view. But some of our works is dedicated and to some base gases, like ammonia, which is also very important for the industry. As for choosing proposed ionic liquids, we are choosing them as most common and some novel. And now we are calculation solubility data for ammonia in different ionic liquids. This led to answer – yes, we can apply current approach to other materials.

2- It will be much better to add some more previous references related to the present work.

We add some reference to our previous works and add some files with them, but concerning the calculation we can’t do this, because this our let say first try to make together our experimental data with some predictions.

We add some text with reference to our related published works in the Introduction part:

“In this regard, the uprising trend of replacing conventional acidic gases separation techniques by the economically viable and energy-effective membrane or membrane assisted technology has been gaining much interest over the last years, also by authors [2-6].”

(2)     Akhmetshina, A.I.; Yanbikov, N.R.; Atlaskin, A.A.; Trubyanov, M.M.; Mechergui, A.; Otvagina, K.V.; Razov, E.N.; Mochalova, A.E.; Vorotyntsev, I.V. Acidic gases separation from gas mixtures on the supported ionic liquid membranes providing the facilitated and solution-diffusion transport mechanisms. Membranes, 2019. 9 (1), art. no. 9.

(3)        Akhmetshina, A.I.; Petukhov, A.N.; Gumerova, O.R.; Vorotyntsev, A.V.; Nyuchev, A.V.; Vorotyntsev, I.V. Solubility of H2S and CO2 in imidazolium-based ionic liquids with bis(2-ethylhexyl) sulfosuccinate anion. Journal of Chemical Thermodynamics, 2019. 130, pp. 173-182. (4)        Atlaskin, A.A.; Kryuchkov, S.S.; Yanbikov, N.R.; Smorodin, K.A.; Petukhov, A.N.; Trubyanov, M.M.; Vorotyntsev, V.M.; Vorotyntsev, I.V. Comprehensive experimental study of acid gases removal process by membrane-assisted gas absorption using imidazolium ionic liquids solutions absorbent. Separation and Purification Technology, 2020. 239, art. no. 116578.(5)        Atlaskin, A.A.; Kryuchkov, S.S.; Smorodin, K.A.; Markov, A.N.; Kazarina, O.V.; Zarubin, D.M.; Atlaskina, M.E.; Vorotyntsev, A.V.; Nyuchev, A.V.; Petukhov, A.N.; Vorotyntsev, I.V. Towards the potential of trihexyltetradecylphosphonium indazolide with aprotic heterocyclic ionic liquid as an efficient absorbent for membrane-assisted gas absorption technique for acid gas removal applications. Separation and Purification Technology, 2021. 257, art. no. 117835.

(6)     Vorotyntsev, V.M.; Drozdov, P.N.; Vorotyntsev I.V.; Murav'ev  D.V. Fine gas purification to remove slightly penetrating impurities using a membrane module with a feed reservoir. Doklady Chemistry, 2006. 411, 243-245.

3 - All the figures looks to blurry. Please replace them with high quality figures.

We add not so blurry figures.

Reviewer 2 Report

Authors used many abbreviations (examples, COSMO-RS....etc.) Please make sure you elaborated before using abbreviations. 

References are not in the format. Please make it consistent.

Author Response

The Authors want to express their appreciation to the Reviewer for the consideration of our article, thorough review, and pointing out the lack of the essential data.

Reviewer

1- Authors used many abbreviations (examples, COSMO-RS....etc.) Please make sure you elaborated before using abbreviations.

We appreciate for this comment and checked the manuscript. It is ok now. Moreover, the list of abbreviations was added in the manuscript.

2- References are not in the format. Please make it consistent..

The reference was proposed in a proper format now.

Best regards,

Authors

15thApril 2020

Reviewer 3 Report

Dear authors,

I would like to reconsider the paper after minor revision.

Please find the comments in the attached pdf file.  

Author Response

Dear authors, I would like to reconsider the paper after minor revision.  Please find the comments in the attached pdf file.   

We appreciate for this work, which you done, and we proposed resubmitted manuscript.

I would recommend the authors to carefully check the text of the Manuscript to track and correct the incorrect sentences and errors/misprints:

  1. Where are the full names or structures of ILs studied? They should be clearly stated at the beginning of the paper. The best way to present the ILs studied is the table with chemical structures of ILs studied, their abbreviation names and some physical properties.

         We add the desired table

  1. Page 3. « Conductor like screening model for real solvents COSMO-RS is a new method for the prediction of thermophysical and chemical properties of fluids and liquid mixtures based on unimolecular». This is the first sentence of the section 2.1

This sentence was deleted.

  1. What is the stucture of anion? Is it docusate-based IL? If so, please point it out clearly.

Yes. It is done.

  1. It would be great if the Authors presented the [empyrr] and [hpyr] ILs as well. In case of absence such experimental data it should be clearly stated in the texth of the Manuscript.

It is really good ideas, but Iin this work, we wanted to reflect the possibility of predicting the properties of IL, so we verified the obtained COSMO-RS data, with the experimental data already obtained by us [mim] [doc] and [bmim] [doc].

  1. Please provide the comparison of predicted values with the experimental ones. The authors can use their earlier work for this (reference [31]).

It was added in the manuscript: “The experimental data on Henry's constants are given also in [31], where its were by 2 and 4 times less for the cations [mim] and [bmim], respectively. Such a large difference is apparently realized due to the fact that the experimental data for these substances are given with a sufficiently high water content of 0.3722 wt% and 0.4835 wt%.”

  1. Please use the consistent units throughout the text: e/nm2 or e/Å2 .

Thank you for the comment. We don it.

  1. From my point of view, this part should be moved to the "2.Theory" section because this is not the result of the work.

We add this part.
